# OpenReview forum: "Is Your Multimodal Language Model Oversensitive to Safe Queries?"
_ICLR.cc/2025/Conference — ICLR 2025 Poster_

### Official Review · Reviewer_UqLu · 2024-10-23

**Soundness:** 3
**Presentation:** 4
**Contribution:** 3
**Rating:** 6
**Confidence:** 5

**Summary:**

This paper introduces a novel benchmark for evaluating the issue of oversensitivity in MLLMs, aiming to balance safety and efficiency. It contributes a dataset in which the image-text inputs are benign but may be mistakenly recognized by MLLMs as violating safety guidelines, leading to a refusal to respond. It provides extensive empirical evaluations and presents several findings. In addition, it designs clever ablation experiments to verify the components that lead to oversensitivity.

**Strengths:**

1. The paper is well-structured with rigorous writing logic, and the experimental design is tightly interconnected. The idea behind the ablation experiments designed at the end is particularly clever.
2. The paper provides intuitive insights into balancing the safety of MLLMs and user efficiency.

**Weaknesses:**

1. First, it is mentioned in the limitations that the dataset size is too small, in terms of cost and resource requirements due to the need for human intervention in the filtering and selection processes. The comprehensiveness and generalizability of the benchmark need improvement.
2. The experimental section only evaluates the rejection rate, which is crucial; however, it lacks an assessment of the balance between safety in question answering and the efficiency of the answers. When using this benchmark to evaluate MLLMs, focusing only on the rejection rate would be too one-sided. The goal is to balance, it would be appropriate to include experiments that assess the correct rejection of harmful queries, the incorrect rejection of harmless queries, the correct response to harmless queries, and the incorrect response to harmful queries.
3. I observed that the prohibitions for some examples related to “Negated Harm” seem somewhat forced subjectively (e.g., figure 20, figure 25).

**Questions:**

1. Regarding the division of the processing steps and the design of the ablation study, I would like to know if these are original contributions of this paper. If so, I would consider giving a higher score.
2. Some typo: It covers a variety of scenariosand serves as a diagnostic suite for assessing oversensitive behaviors. (line84-85); (3). Safety Judgement (line430-431)
3. I believe it is quite necessary, rather than solely focusing on the rejection rate. I will consider increasing the score if you add a set of small experiments during the rebuttal period (weakness two).

---

> ### Author Response · Authors · 2024-11-24
> **Response to R1 UqLu**
>
> Thank you for your careful review! We've addressed your questions in the following sections, and we are happy to discuss any further thoughts you may have!

---

> ### Author Response · Authors · 2024-11-24
>
> **[Concern 1 - Dataset Size]**
>
> *“*First, it is mentioned in the limitations that the dataset size is too small, in terms of cost and resource requirements due to the need for human intervention in the filtering and selection processes. The comprehensiveness and generalizability of the benchmark need improvement.*”*
>
> **[Response]**
>
> Thank you for this point. We indeed have plans to further extend this dataset in future work. Yet as the first paper aiming to raise awareness of the oversensitivity issue, we believe the current dataset size is enough to achieve our goal, as demonstrated in the common reply **[On the sufficient size of the benchmark]**. Below, we offer a brief summary of key points of our demonstration:
>
> - **300 samples are comparable to the benchmark in the safety domain**
>
>     MOSSBench has a comparable size in the safety benchmark field, with 300 unique scenarios (demonstrated by direct comparison with other datasets in common reply).
>
> - **Statistical stability at 300 samples**
>
>     Our experiments demonstrate that results stabilize at 300 samples, ensuring that the dataset size is sufficient to draw reliable conclusions.
>
> - **Focus on common scenarios at the current stage**
>
>     MOSSBench focuses on unique scenarios within each sample and covers tasks and topics frequently encountered in daily life. Highlighting oversensitivity behaviors in these common scenarios is essential for raising awareness.

---

> ### Author Response · Authors · 2024-11-24
>
> **[Concern 2 - Balanced Evaluation]**
>
> *“The experimental section only evaluates the rejection rate, which is crucial; however, it lacks an assessment of the balance between safety in question answering and the efficiency of the answers. When using this benchmark to evaluate MLLMs, focusing only on the rejection rate would be too one-sided. The goal is to balance, it would be appropriate to include experiments that assess the correct rejection of harmful queries, the incorrect rejection of harmless queries, the correct response to harmless queries, and the incorrect response to harmful queries.”*
>
> **[Response]**
>
> Thank you for raising this important consideration. Our study includes two types of samples: oversensitivity samples and contrastive samples, which are specifically designed to probe model behavior across both safe and unsafe queries within similar scenarios.
>
> To address your concern about balancing safety and efficiency, we **updated** our submission with the following metrics in Appendix D.6 (Table 13 and Table 14):
>
> - **Correct rejection of harmful queries and incorrect rejection of harmless queries (Table 13)**
> - **Correct response to harmless queries and incorrect response to harmful queries (Table 14)**
>
> These metrics provide a more comprehensive and balanced evaluation of the model's performance.

---

> ### Author Response · Authors · 2024-11-24
>
> **[Concern 3 - Evaluation Samples]**
>
> *“I observed that the prohibitions for some examples related to “Negated Harm” seem somewhat forced subjectively (e.g., figure 20, figure 25).”*
>
> **[Response]**
>
> Thank you for raising this point and for your thoughtful observation. To address concerns about the subjectivity of evaluation samples, we adopted a majority-vote data filtering approach using Amazon Mechanical Turk (AMT) crowdsourcing, as described in Section 4.1. Annotators were asked, **“Does the request fit appropriately with the context of the image?”** The details of this process are described in Appendix F. This data filtering was designed to incorporate diverse perspectives and ensure that the evaluation samples reflect a broader consensus. By relying on this methodology, we aimed to reduce individual subjectivity and provide a more representative and objective basis for our analysis.
>
> We appreciate your feedback and welcome further suggestions to enhance the clarity and robustness of our evaluations.

---

> ### Author Response · Authors · 2024-11-24
>
> **[Question 1 - Contribution of data processing and ablation study]**
>
> *“Regarding the division of the processing steps and the design of the ablation study, I would like to know if these are original contributions of this paper.”*
>
> **[Response]**
>
> Thank you for your question. The division of processing steps and the design of the ablation study, specifically within the context of oversensitivity, are key original contributions of our work, as outlined below:
>
> **Division of Processing Steps:**
>
> While synthesizing data with the help of LLMs is a common practice, the specific multi-step data processing framework outlined in our paper is our original contribution. The processing steps are specifically crafted to generate diverse oversensitivity scenarios based on the three stimuli types described in Section 3:
>
> - **Candidate Generation**: This step ensures the automatic creation of diverse oversensitivity scenarios.
> - **Candidate Filtering**: This step mitigates the side effects of Candidate Generation, addressing issues such as harmful or unnatural scenarios.
>
> **Design of Ablation Study:**
>
> We recognize that your question may refer to either the system prompts in Section 5 or the reasoning process ablations described in Section 6. Both represent original contributions to our work:
>
> 1. **Ablation on System Prompts:**
>     - Our study is the first to explore how system prompts influence oversensitivity behaviors in MLLMs.
>     - The results demonstrate that safer models exhibit higher oversensitivity, revealing a critical trade-off in model design. This insight provides novel guidance for designing prompts in safety-critical applications.
> 2. **Ablation on Reasoning Processes:**
>     - Inspired by investigation in reasoning[1], we trace refusal behaviors back to the reasoning process, analyzing MLLMs’ descriptions of oversensitivity examples.
>     - By incorporating oracle descriptions and oracle reasoning steps, we demonstrate how different stimuli challenge various stages of the reasoning process.
>     - These findings highlight substantial errors in perception and reasoning, offering insights into model behavior under different oversensitivity scenarios.
>
> [1] Zhang, Yizhe, et al. "How Far Are We from Intelligent Visual Deductive Reasoning?." *arXiv preprint arXiv:2403.04732* (2024)

---

> ### Comment · Reviewer_UqLu · 2024-11-25
> **Reply to Authors' responses**
>
> Thanks for your careful responses and the thorough supplementary experiments. I will update my score accordingly. I hope you can continue to optimize and increase the quality and quantity of the dataset in the future. Good luck!

---

> ### Author Response · Authors · 2024-12-02
>
> Your thorough feedback have been instrumental in elevating the quality of our work. We are particularly grateful for your guidance on the supplementary experiments. Thank you for your supportive comments and constructive recommendations throughout the process!

---

### Official Review · Reviewer_tt7B · 2024-11-02

**Soundness:** 2
**Presentation:** 3
**Contribution:** 3
**Rating:** 6
**Confidence:** 3

**Summary:**

The paper presents a benchmark dataset for testing the possible oversensitivity of Multimodal Large Language Models (Vision + Language), i.e., rejection of harmless queries due to the safety mechanism of MLLM. The authors focused on visual stimuli that can  potentially trigger such systems.
Visual stimuli that cause oversensitivity of MLLM were categorized in three main categories:

1. exaggerated risk, i.e. potential dangerous elements but harmless in the given context;
2. negated harm, i.e. the harmful content is present with a message of discouraging the harm (e.g. a visual negation);
3. counterintuitive interpretation, i.e. elements in benign intentions are misinterpreted.

Authors address two main research questions:
1. how prevalent is oversensitivity in MLLM?
2. what specific failure in MLLM ‘s response process contributes to such behaviors?

To these aims, they developed Multimodal OverSenSitivity Benchmark (MOSSBench), a benchmark of 544 image-text pairs formatted for Visual Question answering (VQA). It utilizes an hybrid method that combine LLM and human inputs for different scenarios and meticulous filtering, then annotated by an AMT annotator. They tested the benchmark on an array of 20 models from 3 categories: proprietary, proprietary from web versions and open sources. They also perform a failure analysis to test at which stage of reasoning the MLLM refuses a query. The results highlight three main findings:

1. Oversensitivity is prevalent on most of the current MLLMs.
2. Safer models are also more oversensitive.
3. Precise type of stimuli triggers more precise stages of the reasoning process.

MOSSBench is collected by a two-step sample generation:
1. candidate generation: using several LLMs (to reduce biases in data) for generating the candidate texts and DALL-E 3 or looking up the web for the images.
2. candidate filtering: human annotators filter the generations to delete possible real harmful content;  9% of the candidates were found to be actually harmful.

Finally, they construct a complementary set of harmful samples by introducing explicit malice into their scenario.

Authors perform both human and automatic evaluation (via GPT4-V), finding a 97.8% agreement among the human annotators.

From the results of such analysis, they highlight 5 main findings:
1. Open source models are less sensitive than proprietary models, except for the models from the GPT-series.
2. Web version of proprietary models shows greater sensitivity with respect to the API version
3. The refusal rate is higher for stimuli of Negated Harm or Counterintuitive interpretation
4. The safer the model, the more oversensitive it is.
5. System prompts influence oversensitive behavior.

For the final analysis about the stage that causes incorrect refusal in MLLMs, the authors divide the response process in three main stages:
1. perception: the model must describe in the correct way the image;
2. intent reasoning: the model formulated hypothesis about the user’s actual intent;
3. safety judgement: the model decide to accept or reject the query, based on the other two stages.

The authors find that most errors occur in the perception stage and different stimuli challenge different stages, i.e. exaggerated risk challenge more the perception stage. Conversely, negated harm and counterintuitive interpretation have most of the errors in intent reasoning and final decision stages.

**Strengths:**

The paper extends a debated topic on LLMs (sometimes called extra safety) to MLLM, clearly focusing on the visual component of such models.

The authors introduce a  useful resource (MOSSBench), identifying a variable dimension about the type of visual stimuli divided in 3 possible categories. They also correlate the Refusal Rate with the actual safetiness of the models, computed by mean of a complement set  of real harmful examples, which is an important comparison to perform.

Also, the post-experiment analysis about the refusal stage is important, in order to understand what the model finds challenging, and the authors show some similarity across the models (same visual stimuli tends to fail for the same “failed” stage).

The evaluation pipeline can be potentially useful for future analysis of safety vs oversensitivity of the models and diagnose the cause of errors.

**Weaknesses:**

In the paper there is a continuous parallelism with human cognition and behavioral disturb that causes similar errors in the humans of what is seen in MLLM. As written, it seems that authors give the MLLM “human cognition” capabilities and similar possible illnesses/disturb. All these could be avoided since they don’t give any contribution to the message of the paper and again, give the hint that MLLMs have something close to human cognition. Instead, a bigger focus on the technical aspects that lead to failures might strenghten the paper.

For instance, example 2 is misleading since the MLLM oversensitivity, in that case, seems totally an error of miscomprehension of the MLLM, and I don’t see any similarity with human cognition.

Moreover, the MLLMs use a non-stochastic decoding (I saw also the impact of temperature on the results). To this matter, results on a single run limit the generalization of the claim. Since the main point of the paper is the resource, a reader would expect to see multiple runs of the experiments on the 3-4 most meaningful models to verify the generalization of the claims, reporting average and standard deviation. This should be done also on the analysis about the error stage.

I am also not totally convinced by the usage of GPT4-V for automatic evaluation (GPT4-V is one of the models used in the experiments), and the paper is missing a comparison between human annotation and LLM annotation

Some minor typos:
- Line 48: “Concretely” 🡪 “concretely”
- Line 84: “scenariosand” 🡪 “scenarios and”
- Check the capitalization of the first letter in the item lists.

**Questions:**

- Have you tried to perform multiple runs on the same model (with fixed hyperparameters)? How much are they coherent with the results reported?

- Can you add a general description of the type of subjects in the images? Can you provide any categorization of the images? I would expect that some categories, due to embedded biases in the model, lead to higher ARR than others (i.e. image with predatory animals vs image of cities).

---

> ### Author Response · Authors · 2024-11-24
> **Response to R2 tt7B**
>
> Thank you for your insightful and constructive feedback. We hope the information below answers your questions, and please feel free to let us know if anything else comes up!

---

> ### Author Response · Authors · 2024-11-24
>
> **[Concern 1 - Parallelism with Human Cognition]**
>
> *“In the paper there is a continuous parallelism with human cognition and behavioral disturb that causes similar errors in the humans of what is seen in MLLM. As written, it seems that authors give the MLLM “human cognition” capabilities and similar possible illnesses/disturb. All these could be avoided since they don’t give any contribution to the message of the paper and again, give the hint that MLLMs have something close to human cognition. Instead, a bigger focus on the technical aspects that lead to failures might strengthen the paper.*
>
> *For instance, example 2 is misleading since the MLLM oversensitivity, in that case, seems totally an error of miscomprehension of the MLLM, and I don’t see any similarity with human cognition.”*
>
> **[Response]**
>
> 1. **Regarding human cognition framing:** We appreciate the reviewer’s thoughtful feedback regarding the framing of parallels between MLLM oversensitivity and human cognitive distortions. Our intent was not to anthropomorphize MLLMs or suggest that they possess human-like cognition or mental states. Rather, these analogies were used as an interpretative tool to better articulate the observed patterns of errors in MLLMs, drawing on established concepts from cognitive psychology to aid understanding. Similar analogies have been used in prior works[1][2]. We recognize that the current framing could lead to misinterpretation. To address this, we have revised the introduction to ensure these parallels are framed more cautiously and explicitly as a metaphorical lens rather than equivalence.
> 2. **Regarding the specific concern about example 2:** We agree that miscomprehension is one driving factor contributing to oversensitivity in MLLMs. In fact, our analysis identifies **miscomprehension as a critical driver of oversensitivity** in several cases, including example 2. Such instances are not mutually exclusive from the broader conceptual framing we propose. Instead, miscomprehension can be viewed as a manifestation of the oversensitivity issue, wherein the model's interpretation of ambiguous inputs leads to exaggerated and premature refusals. To explicitly highlight how miscomprehension contributes to oversensitivity, we also designed an analysis in Section 6, situating it within the larger framework of oversensitivity in MLLMs.
>
> [1] Echterhoff, Jessica, et al. "Cognitive bias in decision-making with llms." *Findings of the Association for Computational Linguistics: EMNLP 2024*. 2024.
>
> [2] Lin, Ruixi, and Hwee Tou Ng. "Mind the biases: Quantifying cognitive biases in language model prompting." *Findings of the Association for Computational Linguistics: ACL 2023*. 2023.

---

> ### Author Response · Authors · 2024-11-24
>
> **[Concern 2 - Multi-Run]**
>
> *“Moreover, the MLLMs use a non-stochastic decoding (I saw also the impact of temperature on the results). To this matter, results on a single run limit the generalization of the claim. Since the main point of the paper is the resource, a reader would expect to see multiple runs of the experiments on the 3-4 most meaningful models to verify the generalization of the claims, reporting average and standard deviation. This should be done also on the analysis about the error stage.“*
>
> **[Response]**
> Thank you for raising this important concern. To address this, we conducted multiple runs of the experiments on the 3-4 most meaningful models to verify the generalization of our claims. The results, including averages and standard deviations, are **updated in Appendix D.1.2** (Table 8: Refusal rate (% ↓) of proprietary MLLMs under multi-run settings). These findings are consistent with the claims presented in Section 5, 6, further supporting their robustness. To address the concern regarding absence of repeated experiments for error analysis, we conducted experiments on an expanded subset of samples **updated in** **Appendix D.3** (Table 11-12: Analysis with more examples). However, due to limited resources and the fact that web-based versions of MLLMs provide very restricted access within specific time periods, we were unable to perform multiple runs to calculate averages and standard deviations across different models.

---

> ### Author Response · Authors · 2024-11-24
>
> **[Concern 3 - Automatic Evaluation Model]**
>
> ”*I am also not totally convinced by the usage of GPT4-V for automatic evaluation (GPT4-V is one of the models used in the experiments), and the paper is missing a comparison between human annotation and LLM annotation*”
>
> **[Response]**
>
> Thanks for bringing up this point. We carefully considered the use of GPT-4V for automatic evaluation and provide the following justifications and updates to address your feedback:
>
> - **Rationale for Using GPT-4V as an Evaluator:**
>     - GPT-4V has became a **common choice** for automatic evaluation in the field due to its scalability, reproducibility, and reduced cost compared to human evaluation [1]. Previous studies [2] have shown that GPT-4V exhibits **a high level of agreement with human evaluations across various tasks**, supporting its reliability as an evaluator. Given these advantages, we employed GPT-4V to provide a reproducible and scalable pipeline for evaluating the target model’s responses.
> - **Comparison Between Human and GPT-4V Evaluations:**
>     - To further validate the alignment between GPT-4V and human annotations, we **updated** the draft with an agreement rate analysis in Appendix C (Table 3 - Evaluation agreement rate between GPT4 evaluation and human evaluation), indicating a strong correspondence between the two, demonstrating the reliability of GPT-4V evaluations in this context.
>
> [1] Liu, Xin, et al. "Mm-safetybench: A benchmark for safety evaluation of multimodal large language models." *European Conference on Computer Vision*. Springer, Cham, 2025
>
> [2] Chen, Dongping, et al. "Mllm-as-a-judge: Assessing multimodal llm-as-a-judge with vision-language benchmark." *arXiv preprint arXiv:2402.04788* (2024).

---

> ### Author Response · Authors · 2024-11-24
>
> **[Question 1 - Analysis with Meta data]**
>
> “*Can you add a general description of the type of subjects in the images? Can you provide any categorization of the images? I would expect that some categories, due to embedded biases in the model, lead to higher ARR than others (i.e. image with predatory animals vs image of cities)*”
>
> **[Response]**
>
> Thank you for your question. General descriptions and metadata for the images are provided in Appendix B.3 (Table 2 - Key statistics of MOSSBench), along with an illustration of harm type distributions in Figure 12. We categorize our samples into critical groups, such as harm types, the presence or absence of humans, and the presence of synthesized versus natural images. These details are outlined in Appendix B.3 and form the basis for our fine-grained analyses of safety categories **updated in Appendix D.2**. Specifically:
>
> - Figure 14 compares ARR for samples of different harm types.
> - Table 9 compares ARR for images featuring humans versus those without.
> - Table 10 compares ARR for natural images versus synthetic images.
>
> Our findings indicate notable differences in ARR across categories, suggesting that certain harm types—likely influenced by embedded biases in the model—may contribute to higher ARR.

---

> > ### Comment · Reviewer_tt7B · 2024-11-27
> > **Thanks for the clarifications**
> >
> > I have read the authors' comments and I thank them for addressing our concerns.

---

> ### Author Response · Authors · 2024-12-02
>
> Your thoughtful input throughout this review process have significantly strengthened our submission. We are pleased to have successfully resolved your concerns. We remains at your disposal should any other questions or comments arise.

---

### Official Review · Reviewer_HYuk · 2024-11-02

**Soundness:** 3
**Presentation:** 4
**Contribution:** 3
**Rating:** 6
**Confidence:** 4

**Summary:**

This work introduces a Multimodal Oversensitivity Benchmark (MOSSBench) to evaluate the oversensitivity of multimodal language models (MLLMs) based on Refusal Rate. The authors categorize the main "stimuli" of oversensitivity into three types: Exaggerated Risk, Negated Harm, and Counterintuitive Interpretation, each explained by an illustrative example. They conduct a thorough and detailed analysis, testing 20 popular MLLMs including both proprietary and open-source models on MOSSBench. They also provide several interesting findings.

**Strengths:**

1. While there is prior work on oversensitivity in language models, this paper is novel as the first to assess oversensitivity specifically in multimodal LLMs.

2. The evaluation is both comprehensive and in-depth, and covering a wide range of MLLMs. The authors provide insightful analyses, such as identifying discrepancies between web and API versions of proprietary models. The design of how different stimuli impact different reasoning stages in MLLMs indicates the novelty of the work and results in several valuable findings.

3. The authors conducted multiple rounds of rigorous controls and reviews on the proposed benchmark, suggesting MOSSBench is in a high quality.

**Weaknesses:**

The dataset is relatively small, with only 300 test cases, which may affect the statistical significance of the findings and limit the impact of this work. A small benchmark size could make it easier for MLLMs to overfit to this evaluation.

**Questions:**

1. Are the ARR results for open-source models (and API versions of proprietary MLLMs) reported after running randomness tests with repeated experiments, such as changing the random seed or decoding strategy?
2. How was the human annotation team? Could you provide details on their occupations or expertise, such as in cognitive science or psychology?

---

> ### Author Response · Authors · 2024-11-24
> **Response to R3 HYuK**
>
> Thank you for your insightful review of our paper! We hope the following responses address your concerns. If you have any further inquiries, please don’t hesitate to reach out!

---

> ### Author Response · Authors · 2024-11-24
>
> **[Concern 1 - Size of the dataset]**
>
> *“The dataset is relatively small, with only 300 test cases, which may affect the statistical significance of the findings and limit the impact of this work. A small benchmark size could make it easier for MLLMs to overfit to this evaluation.”*
>
> **[Response]**
>
> Thank you for your feedback. We also have plans to further extend this dataset in future work. Yet as the first paper aiming to raise awareness of the oversensitivity issue, we believe the current dataset size is enough to achieve our goal, as demonstrated in the common reply **[On the sufficient size of the benchmark]**. Below, we offer a brief summary of key points of our demonstration:
>
> - **300 samples are comparable to benchmark in safety domain**
>
>     MOSSBench has a comparable size in the safety benchmark field, with 300 unique scenarios (demonstrated by direct comparison with other datasets in common reply).
>
> - **Statistical stability at 300 samples**
>
>     Our experiments demonstrate that results stabilize at 300 samples, ensuring that the dataset size is sufficient to draw reliable conclusions.
>
> - **Focus on common scenarios at the current stage**
>
>     MOSSBench focuses on unique scenarios within each sample and covers tasks and topics frequently encountered in daily life. Highlighting oversensitivity behaviors in these common scenarios is essential for raising awareness.

---

> ### Author Response · Authors · 2024-11-24
>
> **[Question 1 - Multi-run of results]**
>
> *“Are the ARR results for open-source models (and API versions of proprietary MLLMs) reported after running randomness tests with repeated experiments, such as changing the random seed or decoding strategy?”*
>
> **[Response to Question 1]**
>
> Thank you for the suggestion. We have incorporated randomness tests with repeated experiments for models in our main experiments. These experiments have been **updated to Appendix D.1**:
>
> 1. Table 6 - Repeated experiments on open-source models with set of random seeds,
> 2. Table 7 - Repeated experiments on open-source models with set of decoding strategies,
> 3. Table 8 - Repeated experiments on API versions of proprietary models.

---

> ### Author Response · Authors · 2024-11-24
>
> **[Question 2 - Information of Annotation Team]**
>
> *“How was the human annotation team? Could you provide details on their occupations or expertise, such as in cognitive science or psychology?”*
>
> **[Response to Question 2]**
>
> Thank you for raising this question. Our study involves two steps of human annotation: **Harmfulness Annotation** and **Sample Metadata and** **Model Response Evaluation Annotation**, each tailored to align the specific goals of the respective tasks.
>
> 1. **Harmfulness Annotation (AMT):**
>
>     As described in Section 4, we used Amazon Mechanical Turk (AMT) to filter and evaluate oversensitivity candidates. The details of the survey are described in Appendix F. While AMT does not provide detailed information about annotators’ occupations or expertise, this approach aligns with our goal of capturing general and representative opinions from a broad, average population.
>
> 2. **Sample Metadata and Model Response Evaluation Annotation (Master’s Students):**
>
>     The evaluation was conducted by two Master’s students in Computer Science, following the detailed evaluation criteria outlined in Appendix C.1.1. This ensured consistency, rigor, and adherence to the methodological framework described in our study.

---

> ### Author Response · Authors · 2024-12-02
>
> As the discussion deadline is approaching, we would like to inquire if you have any further suggestions for improving our manuscript.  Your insights have been instrumental in strengthening our work, and we would be grateful for any further suggestions you may have before the discussion period concludes. Please feel free to share your thoughts whenever convenient.
>
> Thank you for your time and continued consideration of our manuscript.

---

### Official Review · Reviewer_Yuti · 2024-11-04

**Soundness:** 3
**Presentation:** 3
**Contribution:** 3
**Rating:** 6
**Confidence:** 3

**Summary:**

This paper introduces a benchmark, MOSSBench, to systematically evaluate the propensity of multimodal LLMs to refuse to answer harmless queries. The authors find that safer models also tend to refuse more benign queries, that web-based models are more “oversensitive” than API versions, and that particular types of safe queries (i.e. “negated harm” and “counterintuitive interpretation”) engender more refusal than others. The authors perform a deeper analysis of the refusal behavior in a subset of examples, finding that perceptual errors often cause oversensitive behavior.

**Strengths:**

This work tackles a persistent problem in safety-tuned models: their poor ability to properly discern harmless queries from harmful ones. The dataset curation process seems thorough and careful, and the empirical results are compelling

**Weaknesses:**

The primary weakness in this work lies in its rhetorical framing around cognitive distortions. It appears that this comparison is a bit superficial, and does not add much to the scientific contributions of the paper. On the contrary, this framing invites either 1) potentially harmful anthropomorphizing of LLMs, or 2) analogizing people with mental health disorders to LLMs. Though this work does not attempt to use LLMs as models of people with mental health disorders, the current framing may inspire such misguided work in the future.

Additionally, there appears to be an important element of refusal behavior that is absent in the current analysis: the model’s conception of the user submitting the “harmless” query. Prior work [1, 2] has demonstrated that a model’s perception of a user’s attributes importantly modifies their refusal behavior. Understanding person effects seems crucial for understanding the model’s reasons for refusing queries such as those shown in Figs 20, 25, and 30. In all of these cases, the “harmless” query might be rendered harmful if a certain user (i.e., a white supremacist for Fig. 25) is submitting them. In the absence of a user model, perhaps the LLM is merely assuming the worst and responding as such? Including a preliminary investigation of these effects would strengthen the empirical results.

Additionally, some of the most interesting analyses in the paper (Section 6) are performed on a very small number (10) of examples. More examples should be included in the final version of the work.

[1] Zeng, Yi, et al. "How johnny can persuade llms to jailbreak them: Rethinking persuasion to challenge ai safety by humanizing llms." arXiv preprint arXiv:2401.06373 (2024).
[2] Ghandeharioun, Asma, et al. "Who's asking? User personas and the mechanics of latent misalignment." arXiv preprint arXiv:2406.12094 (2024).

**Questions:**

How were these three categories of visual stimuli determined?

---

> ### Author Response · Authors · 2024-11-24
> **Response to R4 Yuti**
>
> We truly appreciate your thoughtful review of our paper. We've provided detailed responses below, and we welcome any additional questions or feedback you may have:

---

> ### Author Response · Authors · 2024-11-24
>
> **[Concern 1 - Framing around Cognitive Distortions]**
>
> *“The primary weakness in this work lies in its rhetorical framing around cognitive distortions. It appears that this comparison is a bit superficial, and does not add much to the scientific contributions of the paper. On the contrary, this framing invites either 1) potentially harmful anthropomorphizing of LLMs, or 2) analogizing people with mental health disorders to LLMs. Though this work does not attempt to use LLMs as models of people with mental health disorders, the current framing may inspire such misguided work in the future.”*
>
> **[Response]**
>
> Thank you for your thoughtful feedback on the framing of our work. Our comparison to cognitive distortions was intended as a conceptual tool to illustrate the oversensitivity phenomenon in MLLMs, rather than to anthropomorphize LLM behaviors or draw parallels to human mental health conditions. Similar analogies have been used in prior works[1][2]. We also recognize the potential for misinterpretation and have taken several steps to address this in the revised version of our paper.
>
> **To address your concern, we have make the following adjustments:**
>
> 1. **Clarified the purpose of the analogy:** We **revised the Introduction** to explicitly state that the reference to cognitive distortions is solely a framing device to explain oversensitivity and is not meant to equate LLM behaviors with human mental processes or mental health conditions.
> 2. **Re-focused the framing:** We **adjusted the language** throughout the paper to emphasize our work’s technical and empirical contributions, ensuring that the focus remains on the observed oversensitivity phenomenon and the systematic analysis we conducted. We aim to minimize the risk of inappropriately anthropomorphizing LLMs and analogizing people with mental health disorders to LLMs.
>
> We appreciate your feedback and welcome further suggestions to improve the clarity and accessibility of our work.
>
> [1] Echterhoff, Jessica, et al. "Cognitive bias in decision-making with llms." *Findings of the Association for Computational Linguistics: EMNLP 2024*. 2024.
>
> [2] Lin, Ruixi, and Hwee Tou Ng. "Mind the biases: Quantifying cognitive biases in language model prompting." *Findings of the Association for Computational Linguistics: ACL 2023*. 2023.

---

> ### Author Response · Authors · 2024-11-24
>
> **[Concern 2 - User’s Attributes in Analysis]**
>
> *“Additionally, there appears to be an important element of refusal behavior that is absent in the current analysis: the model’s conception of the user submitting the “harmless” query. Prior work [1, 2] has demonstrated that a model’s perception of a user’s attributes importantly modifies their refusal behavior. Understanding person effects seems crucial for understanding the model’s reasons for refusing queries such as those shown in Figs 20, 25, and 30. In all of these cases, the “harmless” query might be rendered harmful if a certain user (i.e., a white supremacist for Fig. 25) is submitting them. In the absence of a user model, perhaps the LLM is merely assuming the worst and responding as such? Including a preliminary investigation of these effects would strengthen the empirical results.”*
>
> **[Response]**
>
> Thank you for raising this point about the role of user attributes in shaping model behavior. While we recognize that understanding how a model perceives a user’s attributes is a valuable direction for research and could provide additional insights into oversensitivity behavior, we feel that this investigation falls outside the scope of our current study.
>
> **The primary objective of MOSSBench:**
>
> The **primary objective** of MOSSBench is to raise awareness about the issue of oversensitivity in MLLMs. To achieve this, the scope of the study was adopted to focus on analyzing oversensitivity as a **general behavior** exhibited by MLLMs. This focus allows us to clearly highlight the prevalence and generality of oversensitivity across models, avoiding additional complexities. To achieve this, our data collection process and verification of query benigness were conducted independently of any presumed user assumptions, as discussed in Section 4.1. This ensures our analysis is self-contained, and grounded in **general perceptions of harmlessness**.
>
> **Explanation of cases above:**
>
> Specifically, the cases highlighted in Figs. 20, 25, and 30 demonstrate that oversensitivity arises when the input queries are **objectively benign.** While it is possible that the model might be “assuming the worst” about the user, we believe that such behavior might reflect a broader issue with the model’s reasoning and safety mechanisms, which we aim to investigate and bring to light.
>
> We recognize that incorporating user attributes could further enrich the analysis, and we see this as a natural extension of our work for **future research**. In this case, introducing user modeling requires additional data, ethical considerations, and methodological rigor, which are beyond the scope of our current study.
>
> We appreciate your thoughtful feedback and welcome further discussions to improve our work. Thank you for helping us refine our approach and presentation.

---

> ### Author Response · Authors · 2024-11-24
>
> **[Concern 3 - Analysis with small number of examples]**
>
> *“Additionally, some of the most interesting analyses in the paper (Section 6) are performed on a very small number (10) of examples. More examples should be included in the final version of the work.”*
>
> **[Response]**
>
> Thank you for raising this concern. To address this, we have expanded our analysis to include (10->30) examples per category, resulting in a total of (30->90) samples. The updated results have been **updated as a revision** in Appendix D.3 (Table 11 - Perception investigation with more instances; Table 12 - Tracing MLLMs’ behavior with more instances) of the paper. The results are consistent to those claims in Section 6.

---

> ### Author Response · Authors · 2024-11-24
>
> **[Question 1 - Determination on Three Oversensitivity Types]**
>
> *“How were these three categories of visual stimuli determined?”*
>
> **[Response]**
>
> Thank you for the question. The three categories of visual stimuli were identified through an initial exploratory analysis. We focus on these three stimuli types based on the following reasons:
>
> **Focus on Three Stimuli Types**:
> a) **Relevance**: Most discovered oversensitivity-triggering cases fall into these categories
> b) **Prevalence:** They are generic enough to affect most existing MLLMs. This focus allows us to solidify our findings and understanding, laying the groundwork for future exploration.
> Because of these reasons, instead of exhaustively listing all possible scenarios, we double down our efforts on solidifying the findings and understanding of these behaviors, so that we can lay a solid ground for future explorations.

---

> > ### Comment · Reviewer_Yuti · 2024-11-25
> >
> > Thank you for addressing my concerns, especially with respect to the framing of the paper. I have adjusted my score to reflect these changes.

---

> ### Author Response · Authors · 2024-12-02
>
> We sincerely appreciate your detailed feedback and active participation throughout the review process. We are delighted that we were able to address your concerns successfully. Please do reach out if you have any further concern or have additional comments to share.

---

### Author Response · Authors · 2024-11-24
**Common reply to all reviewers**

Dear reviewers,

We sincerely thank you all for your time and effort in reviewing our paper on MLLM oversensitivity. Your insightful comments and constructive feedback have been invaluable in helping us improve our work. We are particularly encouraged by the positive feedback we received on the **novel topic and research question** (R2, R3), **useful resources** (R1, R2, R3, R4), **well-structured** **presentation** (R1), **comprehensive evaluation** (R1, R3), and **insightful results** (R1, R2, R3, R4).

**We also greatly value the concerns raised, particularly regarding the framing around cognitive distortions and the dataset size of MOSSBench**. These points have significantly helped refine our paper. Below, we summarize our clarifications to major concerns and outline the adjustments made to the manuscript.

---

> ### Author Response · Authors · 2024-11-24
>
> **[On the framing around cognitive distortions]**
>
> We understand the reviewers’ concerns regarding the framing of oversensitivity in MLLMs through the lens of cognitive distortions. Our original intention was to provide an intuitive, conceptual tool to communicate the nature of oversensitivity, drawing inspiration from analogous concepts in cognitive psychology. This approach was **not** intended to anthropomorphize LLM behaviors, equate them to human cognition, or make associations with humans with mental health conditions. Prior works [1][2] also employ similar analogies, indicating such framings can effectively bridge complex technical findings to more accessible conceptual frameworks without undermining scientific accuracy.
>
> We also recognize that such framing can be misinterpreted and may unintentionally detract from the scientific focus of our work.
>
> To address these concerns, we have made the following revisions to our manuscript:
>
> 1. **Clarification of Intent:** We have revised the Section Introduction to explicitly state that references to cognitive distortions serve solely as a metaphorical framing device to articulate oversensitivity patterns. These analogies are not indicative of equivalence between LLM behaviors and human mental states.
> 2. **Minimizing Anthropomorphizing Language:** Throughout the paper, we have adjusted the language to emphasize the technical and empirical contributions of our work, reducing reliance on human-centric metaphors. This ensures that the discussion remains grounded in the systematic analysis of MLLM behavior.

---

> ### Author Response · Authors · 2024-11-24
>
> **[On the sufficient size of the benchmark]**
>
> As **the first to explore and raise awareness of oversensitivity** in MLLMs, our goal is to highlight the most general and widely applicable cases.
>
> The current dataset size is enough to achieve the goal described above for the following reasons:
>
> - **Existing safety benchmarks also contain around 100 examples**
>
>     Most existing MLLM safety benchmarks consist of around 100 examples, with some having even smaller effective sizes when duplicate scenarios are removed. As shown in the table below, MOSSBench is comparable in size to other benchmarks in the field.
>
>     Table 1. *Size and number of unique scenarios of safety datasets. MOSSBench shows a high number of unique scenarios.*
>
>     | Model | Size | Unique Scenario |
>     | --- | --- | --- |
>     | HarmBench[3] | 510 | 510 |
>     | AdvBench[4] | 520 | 58 |
>     | SIUO[5] | 168 | 168 |
>     | Liu et al.[6] | 40 | 40 |
>     | Shen et al.[7] | 390 | 390 |
>     | MaliciousInstruct [8] | 100 | 100 |
>     | Zeng et al. [9] | 42 | 42 |
>     | Dent et al.[10] | 50 | 50 |
>     | Shah et al.[11] | 43 | 43 |
>     | **MossBench (Ours)** | **300** | **300** |
> - **We stopped at 300 because the results are statistically stabilized**
>
>     We conducted an additional experiment to demonstrate that 300 samples are sufficient to arrive at a stable and reliable conclusion. We downsampled our benchmark to 90 samples, evenly distributed across three categories (30 samples per category). In Table 2, we compare the evaluation results between the full 300-sample MOSSBench and the downsampled 90-sample subset. They do not show significant differences in the outcomes, indicating that the current dataset size does not affect the effectiveness of the evaluation.
>
>     Table 2. *Refusal rate (% ↓) of MLLMs on the 90-sample subset and the whole set of MOSSBench by GPT evaluation.* *The reject rate does not differ much between using 90 samples and 300 samples. So the 300 samples are more than enough to arrive at a stable and reliable conclusion.*
>
>     | Model | 90 Samples | 300 Samples |
>     | --- | --- | --- |
>     | GPT-4o | 5.55 | 6.33 |
>     | Claude-3 opus | 32.22 | 34.67 |
>     | Gemini-1.5 | 30.00 | 29.33 |
>     | Llava-1.5-7b | 7.78 | 12.33 |
>     | Qwen-VL-Chat | 21.11 | 21.67 |
>     | InternLM | 22.22 | 17.67 |
> - **Size does not necessarily indicate a lack of diversity**
>
>     Data samples in MOSSBench are designed with diversity in mind. (1) Each sample represents a unique scenario. (2) as demonstrated in Table 3 - Statistics of MOSSBench，MOSSBench encompasses a wide range of oversensitivity stimuli with varying potential harm and **addresses tasks and topics frequently encountered in daily life**. Rather than cherry-picking edge cases, MOSSBench reveals the severity of this oversensitivity in common application scenarios of MLLMs, and meanwhile providing insights into less frequent situations.

---

> ### Author Response · Authors · 2024-11-24
>
> Below are the **highlighted changes to the revision (marked with blue):**
>
> 1. [TEXT] + Revised Introduction to avoid misinterpretation between oversensitivity and cognitive distortion  (R2, R4)
> 2. [EXP] + Table 3: An evaluation of agreement rate between GPT-4V evaluation and human evaluation (R2)
> 3. [EXP] + Table 6-8: Multi-run experiments of MLLMs with respective configurations (R2, R3)
> 4. [EXP] + Table 9-10: Fine-grained analysis of proprietary MLLMs with meta data (R2)
> 5. [EXP] + Table 11-12: Analysis with more examples (R2, R4)
> 6. [EXP] + Table 14-15: Comprehensive and balanced metrics of MLLMs’ safety behavior (R1）
>
> We hope our response can help you in finalizing the ratings of our paper. If you have any other questions, please feel free to reply back, and we will answer them asap!
>
> Sincerely,
> The Authors of Submission 8877
>
> [1] Echterhoff, Jessica, et al. "Cognitive bias in decision-making with llms." *Findings of the Association for Computational Linguistics: EMNLP 2024*. 2024.
>
> [2] Lin, Ruixi, and Hwee Tou Ng. "Mind the biases: Quantifying cognitive biases in language model prompting." *Findings of the Association for Computational Linguistics: ACL 2023*. 2023.
>
> [3] Mazeika, Mantas, et al. "Harmbench: A standardized evaluation framework for automated red teaming and robust refusal." *arXiv preprint arXiv:2402.04249* (2024).
>
> [4] Zou, Andy, et al. "Universal and transferable adversarial attacks on aligned language models." *arXiv preprint arXiv:2307.15043* (2023).
>
> [5] Wang, Siyin, et al. "Cross-Modality Safety Alignment." *arXiv preprint arXiv:2406.15279* (2024).
>
> [6] Liu, Yi, et al. "Jailbreaking chatgpt via prompt engineering: An empirical study." *arXiv preprint arXiv:2305.13860* (2023).
>
> [7] Shen, Xinyue, et al. "" do anything now": Characterizing and evaluating in-the-wild jailbreak prompts on large language models." *arXiv preprint arXiv:2308.03825* (2023).
>
> [8] Huang, Yangsibo, et al. "Catastrophic jailbreak of open-source llms via exploiting generation." *arXiv preprint arXiv:2310.06987* (2023).
>
> [9] Zeng, Yi, et al. "How johnny can persuade llms to jailbreak them: Rethinking persuasion to challenge ai safety by humanizing llms." *arXiv preprint arXiv:2401.06373* (2024).
>
> [10] Deng, Gelei, et al. "Masterkey: Automated jailbreaking of large language model chatbots." *Proc. ISOC NDSS*. 2024.
>
> [11] Shah, Rusheb, et al. "Scalable and transferable black-box jailbreaks for language models via persona modulation." *arXiv preprint arXiv:2311.03348* (2023).

---

### Meta-Review · Area_Chair_bsep · 2024-12-19

**Metareview:**

Overall, the consensus is that this paper is marginally above threshold.  Initially, reviewers found the paper weak in several aspects including that the contribution was superficial, that "refusal behavior" wa snot adequately modeled and that the dataset was relatively small containing only 300 examples.  Two reviewers were uncomfortable with the "anthropomorphism" of LLMs, equating/using a parallel analogy of their behavior with human behavior.  Another concern was the exclusive use of GPT4 for evaluation and no human evaluation.

I am recommending acceptance based on the authors successful defense of their work during the rebuttal period, but I agree that the paper could be stronger.

**Additional Comments On Reviewer Discussion:**

The authors were very responsive in the rebuttal phase and two reviewers moved from marginally reject to marginally accept.

Currently all reviewers scores are at 6, marginally accept.

I believe that the reviewing process worked as it should with the authors being given a chance to clarify their contribution and reviewers reconsidering their position and positively adjusting their scores.

---

### Decision · Program_Chairs · 2025-01-22

Accept (Poster)